# Novel Method for Early Prediction of Clinically Significant Drug–Drug Interactions with a Machine Learning Algorithm Based on Risk Matrix Analysis in the NICU

**DOI:** 10.3390/jcm11164715

**Published:** 2022-08-12

**Authors:** Nadir Yalçın, Merve Kaşıkcı, Hasan Tolga Çelik, Karel Allegaert, Kutay Demirkan, Şule Yiğit, Murat Yurdakök

**Affiliations:** 1Department of Clinical Pharmacy, Faculty of Pharmacy, Hacettepe University, Ankara 06100, Turkey; 2Department of Biostatistics, Faculty of Medicine, Hacettepe University, Ankara 06100, Turkey; 3Division of Neonatology, Department of Child Health and Diseases, Faculty of Medicine, Hacettepe University, Ankara 06100, Turkey; 4Department of Pharmaceutical and Pharmacological Sciences, KU Leuven, 3000 Leuven, Belgium; 5Department of Development and Regeneration, KU Leuven, 3000 Leuven, Belgium; 6Department of Hospital Pharmacy, Erasmus Medical Center, 3000 GA Rotterdam, The Netherlands

**Keywords:** drug–drug interactions, machine learning, neonatal intensive care unit, adverse drug reactions

## Abstract

*Aims:* Evidence for drug–drug interactions (DDIs) that may cause age-dependent differences in the incidence and severity of adverse drug reactions (ADRs) in newborns is sparse. We aimed to develop machine learning (ML) algorithms that predict DDI presence by integrating each DDI, which is objectively evaluated with the scales in a risk matrix (probability + severity). *Methods:* This double-center, prospective randomized cohort study included neonates admitted to the neonatal intensive care unit in a tertiary referral hospital during the 17-month study period. Drugs were classified by the Anatomical Therapeutic Chemical (ATC) classification and assessed for potential and clinically relevant DDIs to risk analyses with the Drug Interaction Probability Scale (DIPS, causal probability) and the Lexicomp^®^ DDI (severity) database. *Results:* A total of 412 neonates (median (interquartile range) gestational age of 37 (4) weeks) were included with 32,925 patient days, 131 different medications, and 11,908 medication orders. Overall, at least one potential DDI was observed in 125 (30.4%) of the patients (2.6 potential DDI/patient). A total of 38 of these 125 patients had clinically relevant DDIs causing adverse drug reactions (2.0 clinical DDI/patient). The vast majority of these DDIs (90.66%) were assessed to be at moderate risk. The performance of the ML algorithms that predicts of the presence of relevant DDI was as follows: accuracy 0.944 (95% CI 0.888–0.972), sensitivity 0.892 (95% CI 0.769–0.962), F1 score 0.904, and AUC 0.929 (95% CI 0.874–0.983). *Conclusions:* In clinical practice, it is expected that optimization in treatment can be achieved with the implementation of this high-performance web tool, created to predict DDIs before they occur with a newborn-centered approach.

## 1. Introduction

Undesirable effects that occur in a potential or clinically relevant way with the concurrent use of two or more drugs are drug–drug interaction (DDI) and adverse drug reactions (ADR) [1]. In the broadest sense, a DDI occurs whenever one drug affects the pharmacokinetics (PK), pharmacodynamics (PD), efficacy, or toxicity of another drug depending on various factors such as drug-related (such as the mechanism of action, route of administration, dose, dose interval, duration of treatment, dosing times) and patient-related (such as diagnosis, polypharmacy, pharmacogenetics, length of hospital stays) [2,3,4] factors. DDIs often lead to increased healthcare costs, morbidity, and mortality, originating from 2.5 to 4.4% of ADRs and 3 to 5% of all inpatient medication errors [5,6].

Many DDIs in neonatal intensive care unit (NICU) patients can remain unrecognized by considering these various factors as well as the workload of the health care professionals. Neonates, particularly admitted to the NICU, have increased the severity of DDIs to result in more common/severe ADR compared to other populations due to physiological/organ immaturity, congenital diseases, birth-related complications, and significant differences in PKs such as extravascular total body water, immature renal/hepatic functions, plasma protein concentrations, blood–brain barrier permeability [7,8]. As a recent illustration of this complexity of DDI in neonates, Salerno et al. explored the impact of co-administration of fluconazole on sildenafil disposition, including the PD-relevant metabolite (N-desmethyl sildenafil). Interestingly, the AUC fold change in adults (2.11-fold) was different in infants (2.82-fold), necessitating a dose reduction of about 60% to attain similar exposure [9].

With the digitalization of health and medicine and the widespread use of electronic health records (EHR), healthcare professionals have begun to adopt the latest methodologies of artificial intelligence (AI). Machine learning (ML) algorithms, a subtype of AI, can act as a kind of co-pilot and predict DDIs before they occur with a patient-centered approach. Due to the lack of comprehensive experimental data for neonates, high study cost, and long experimental duration, the use of computational prediction and DDIs assessment is an encouraging strategy to improve precision medicine: recognize the cases at higher risk to mitigate risks. [10].

Although software DDI checkers for adults are widely available, most have limited clinical utility, especially for neonates. In this context, it was aimed to develop ML algorithms and web tool that predict high-performance DDI by integrating each DDI, which is objectively evaluated with the scales in risk matrix (probability + severity) (http://www.softmed.hacettepe.edu.tr/NEO-DEER_Drug_Interaction/ (accessed on 7 August 2022)).

## 2. Methods

### 2.1. Study Design and Population

Newborns (postnatal age between 0 and 28 days), patients admitted to the NICU for at least 24 h, and patients who received at least one systemic drug during their hospital stay were included in this double-center and prospective randomized cohort study from February 2020 to June 2021. The newborns with hepatic or renal impairment excluded in the study. The Institutional Review Board of Hacettepe University ethical approved this study and written informed consent was obtained from each parent/legal guardian of the participant (decision no. 2020/11-21). This study registered with the ClinicalTrials.gov (accessed on 7 August 2022) (NCT04899960).

### 2.2. Data Acquisition

Patients’ follow-up was performed daily to acquire the clinical status via a comprehensive assessment by the multidisciplinary team including physicians, nurses, and a clinical pharmacist. Demographical, clinical, and drug administration data were obtained from routine follow-up for each patient. International Classification of Diseases 10th Revision (ICD-10) codes for diagnoses, Anatomical Therapeutic Chemical (ATC) codes for categorization of drugs were used for all patients.

### 2.3. Causal Probability, Severity, a Risk Matrix Development of DDIs

Potential DDIs with all drugs prescribed simultaneously in each NICU patient until discharge was prospectively determined using the Lexicomp^®^ DDI database, clinical and laboratory findings by the clinical pharmacist. The inhibitor/inductor and substrate (victim) drugs, mechanism of DDIs, ADRs of clinically relevant DDIs, and duration of exposure (days) were prospectively registered.

The Drug Interaction Probability Scale (DIPS) was used to determine the *causal probability* for each potential DDI. The DIPS consists of 10 questions and each question is answered as ‘Yes’, ‘No’ or ‘Unknown or Not Applicable (NA)’, DDIs are categorized as >8 points ‘highly probable’, 5–8 points ‘probable’, 2–4 points ‘possible’, and <2 points ‘doubtful’ [11]. By consensus, all DDIs except the ‘doubtful’ were considered clinically significant (any score ≥ 2). In this study, the probability categories were numbered between 1 (doubtful) to 4 (highly probable).

The Lexicomp^®^ DDI database was used to determine the potential *severity* of each DDI. According to the Lexicomp^®^ database, DDIs are rated as X (avoid combination), D (consider therapy modification), C (monitor therapy), B (no action needed), and A (no known interaction) [12]. In this study, the severity categories obtained were numbered between 1 (A = no known interaction) and 5 (X = avoid combination).

These categories are placed in rows (severity, 1–4) and columns (probability, 1–5) in the *risk matrix,* which consists of risk scores obtained by multiplying severity and probability values. In this risk matrix, the risk category was obtained for each DDI as low (white), moderate (light gray), and high (dark gray) risk. This risk matrix created was approved by the consensus of the clinicians involved in the study.

### 2.4. Establishment, Optimization, and Validation of Random Forest Model

Primarily, statistically significant correlations and differences were examined among all general and prescription information (as input variables), and the presence of DDI(s) during hospitalization (as output variables) by univariate analyses. Input variables that were found to be significant according to univariate analysis were chosen as independent variables (*p* < 0.05). Secondly, the data set containing the dependent and independent variables was randomly divided into train sets (70%) for obtain models and test set (30%) for obtaining model performance. Since the 10-fold cross-validation method separates the train data into train and validation sets, a separate validation set was not used when dividing the data set. Elastic net, random forest (RF), and support vector machine (SVM) with different kernel functions were used to compare model performances. The highest performance was provided by RF and the study was analyzed with RF. The accuracy, sensitivity (recall), specificity, positive predictive value (precision-PPV), negative predictive value (NPV), F1 score, and area under ROC curve (AUC) were used as performance measures in classification models to compare the performance of the models. A high-performance model requires these measurements of 0.70 and above.

Accuracy is the ratio of correctly classified samples to the total number of samples. Sensitivity and specificity are the ability of a model to correctly identify positive and negative samples, respectively. PPV is an indicator of how many of the samples classified as positive by the model are actually positive. NPV is an indicator of how many of the samples classified as negative by the model are actually negative. F1 score is harmonic mean of precision and recall. Lastly, AUC is an indicator of how well the classes are separated from each other according to the model obtained.

The model performances were compared after parameter optimization to avoid overfitting with the tuneLength argument in the Classification and Regression Training (caret) package [13]. Variable importance plots were used in the study to show the importance order of the variables used in the prediction models.

Finally, data were collected prospectively to examine the predictive validity of the ML-based model in a different hospital (Erasmus Medical Centre Sophia Children’s Hospital) and country (The Netherlands) by the same web tool user (clinical pharmacist) to ensure quality standard.

### 2.5. Statistical Analysis

For predictive models based on ML, it is not possible to measure the effect size as in hypothesis testing. Instead of calculating the sample size according to the effect size of a certain power level, rules of thumb such as taking 10 or 20 times the number of independent variables as the sample size can be applied. In this study, it was planned to have a maximum of 20 independent variables in the final models, so the minimum sample size was determined as 400 patients, with 20 observations per variable [14]. Continuous variables were defined as the mean (standard deviation, SD) and median (range). The normality of continuous variables was tested using the Shapiro–Wilk test. Categorical variables were defined as percentages and were compared using the χ^2^ test. Univariate analysis was carried out in *IBM SPSS Statistics Version 23* software. For all tests, *p* < 0.05 was considered statistically significant. All ML analyses were performed, using the open-source software R (version 3.6.3, http://www.rproject.org (accessed on 7 August 2022)). In terms of reproducibility, the seed number was set at 1234. Caret [13] package was used as the primary package for model training, 10-fold cross-validation, and variable importance. pROC [15], precrec [16], and ggplot2 [17] R packages were used for obtaining the ROC curve. The quantitative features were normalized before training the models.

## 3. Results

### 3.1. Clinical Characteristics

During the study period, 468 newborns were admitted to the 22-bed capacity NICU in a tertiary referral hospital. Fifty-six patients were excluded because of non-survival (n = 21, 4.5%) or not receiving any systemic drug (n = 35, 7.4%). Therefore, 412 patients were included in the study: 232 (56.3%) were males, 177 (43%) were preterm births, and 172 (41.7%) were low birth weight (<2500 g). According to the numeric variables, the median (IQR) postnatal age (PNA) was 1 (1) day and the median (IQR) length of hospital stay (LOS) was 8 (11) days. General and postnatal information about patients is given in Table 1.

A total of 412 NICU patients (5.53 drugs/patient/day) to whom 2280 drugs were prescribed in 32,925 patient days and 11,908 medication orders (28.9 order/patient) were prescribed with the computerized physician order entry (CPOE) system were prospectively examined from prescribing to the follow-up process. The median (range) values of the total number of drugs and anti-infectives used during the hospitalization period were 3 (0–29) and 2 (0–9) respectively. According to the ATC, the most frequently prescribed drugs in these orders were anti-infective (38.82%), alimentary tract and metabolism (32.89%), and nervous system (8.07%) drugs. In the study period, a total of 131 different drugs were prescribed. The most commonly used of these agents were intravenous fluids (12.06%), gentamicin (8.03%), and ampicillin (7.81%). The rate of anti-infectives among the total number of drugs prescribed was 39.96% (Appendix A).

### 3.2. Characteristics of Potential and Observed DDIs: Incidence and Severity

At least one potential DDI was observed in 125 (30.4%) of the patients included in the study. The total number of potential DDIs detected was 328 (2.6 potential DDI/patient, range 1–15). Of these patients, 66 (52.8%) had 1 DDI, 15 (12.0%) 2 DDIs, 19 (15.2%) 3 DDIs, 25 (20.0%) 4 or more DDIs.

Of 125 patients with potential DDIs, 38 (30.3%) had clinically relevant DDIs known to cause ADR were identified. The total number of clinically relevant DDIs observed in these 38 patients was 75 (2.0 clinically relevant DDI/patient, range: 1–5) (Figure 1). The vast majority (65/75, 99.6 %) of observed clinically relevant DDIs were found to be of moderate risk. Low- and high-risk clinically relevant DDIs were seen in 3 and 4 patients, respectively. According to the risk matrix, while the mean risk score is 10.3 in patients with potential DDIs, this score increases to 21.1 in patients with only clinically relevant DDIs.

The most common clinically relevant DDI observed in patients was between vancomycin and amikacin (17.3%). As a result of this DDI, the mean creatinine was above the upper level to the baseline on the 17th day of this combination. It was recognized that this DDI was determined as a ‘possible’ probability and C (monitoring) severity level. As a result, when the probability and severity data were placed in the risk matrix, it was seen that the DDI was ‘moderate’ risk. DDIs with clinical findings identified as high risk were amiodarone–flecainide, caffeine–adenosine, midazolam–fentanyl, and linezolid–salbutamol (Table 2).

When the probability and severity analysis of all potential DDIs were evaluated separately, ‘doubtful’ probability (77.44%) and moderate (C = monitor therapy) severity (79.88%) of DDIs were most commonly observed in the risk matrix (Table 3).

### 3.3. Development and Optimization of a Model to Predict the Presence of Clinically Relevant DDI

The parameters that have the highest correlation with DDIs and are included in the model were: the total number of anti-infectives, total number of drugs, nervous system drugs, cardiovascular system drugs, respiratory system drugs, and anti-infectives. When considering the importance of the parameters included in the model, it was seen that the most effective variable in predicting the DDI is the total number of anti-infectives (Figure 2).

The obtained model showed very high performance in predicting the presence of DDI. Performance measurements of the model were as follows: accuracy 0.944 (95% CI 0.888–0.972), sensitivity 0.892 (95% CI 0.769–0.962), selectivity 0.966 (95% CI 0.913–0.991), PPV 0.917 (95% CI 0.812–0.966), NPV 0.955 (95% CI) 0.906–0.979), F1 score 0.904, and AUC 0.929 (95% CI 0.874–0.983). The high AUC indicates that the model predicting the presence of DDI correctly classified 92.9% of the patients (Figure 3).

Data were collected prospectively to examine the predictive validity of the model. In total, a sample of 51 NICU patients was reached and 15.7% (n = 8) had observed DDI. The model correctly classified 92% of them. Sensitivity and NPV were obtained as 0.75. Sensitivity and PPV were obtained as 0.92. Similar to the results of the test data set, prospective data set results had high sensitivity and PPV.

## 4. Discussion

This study confirmed that the presence of ADRs (causal probability, DIPS) and risk category of each DDI (severity, Lexicomp^®^ DDI database) varies between patients. Related to this variability, it was shown that the presence of DDIs can be predicted in neonates by using ML algorithms that show high prediction performance in such complex models.

It is estimated that >70% of neonates in the NICU are exposed to DDIs [18]. In our study, a total of 328 potential DDIs were detected in 30.4% of the patients included. More than half of the patients had only one potential DDI during hospitalization. In 30.3% of these patients with potential DDIs, clinically relevant DDIs were determined by an objective DIPS. Looking at the broader picture, a potential DDI was detected in approximately a third of the patients included in the study, and a clinically relevant DDI was detected in a third of these patients (9.2% of the study population). Similarly, Choi et al. identified clinically relevant DDIs in 16 (10.1%) PICU patients [19]. When putting our observations on DDI incidence (potential 30%, clinically relevant 9%) into perspective, other cohorts reported potential DDI incidences of 70, 13.2, or 66.2% [7,20,21].

The DIPS and Lexicomp^®^ database are mostly used in adults in clinical practices and studies. Although there is limited research on the implementation of the DIPS [22,23] and Lexicomp^®^ DDI database [24] in children, there is no study for its implementation in neonates in the current literature. To the best of our knowledge, this is the first study to use these measurements in neonates. Related to the causal probability and severity of each potential DDI detected with the risk analysis matrix (Table 3), 77.44% were ‘doubtful’ according to the DIPS, and 79.88% were in ‘category C = monitor’ according to the Lexicomp^®^ database. In a study evaluating the prevalence of potential DDIs in the NICU, 61.4% of these were in category C [21]. In a single-center retrospective study evaluating the causal probability of DDIs, it was determined that 54.5% of clinically relevant DDIs were ‘probable’ [19]. In our study, clinically relevant DDIs were determined as ‘probable’ at a lower rate (31.57 %). Accordingly, it is understood that the vast majority of DDIs are potentially ongoing, and monitoring is sufficient for these DDIs without the need for any intervention such as drug change, dose change, or drug discontinuation.

While 29–50% of potential DDIs were classified as major in two studies conducted in the NICU, only 12.5% of DDIs were classified in the D or X category in our study [7,25]. In another study conducted in the NICU, 37.5% of potential DDIs were determined as severe or contraindicated [20].

When the DDIs were examined on causal probability (DIPS), only three potential DDIs were identified as ‘highly probable’. These DDIs were observed for amiodarone–flecainide (day 3 of use), midazolam–fentanyl (day 3 of use), and ciprofloxacin–phenytoin (day 5 of use) (Table 2). For a full overview of the ADRs observed, refer to Table 2.

In the literature, the use of CPOE itself has not been associated with a significant decrease in the rate of DDIs [4]. Therefore, there is a need for further development of a clinical decision support system (CDSS) with ML algorithms within CPOE systems. Most of the alerts generated by the legacy CDSS were related to DDIs and dosages [26]. Although there are theoretical and review ML studies on DDI extraction from the biomedical literature [27], DrugBank and other databases [28,29], bioinformatics algorithms to predict DDI [30], and clinical safety DDI information retrieval [31], there are no real-life studies that reflect clinical practice in neonates.

In our study, due to the balanced distribution of the patients with and without the DDI, the high-performance model that predicts whether a DDI will occur in a patient with ML algorithms has been designed successfully. According to this model, the most important variables used in the prediction were, respectively: the total number of anti-infective drugs; the total number of drugs; and nervous system, endocrine system, cardiovascular system, respiratory system, and anti-infective drugs (AUC: 0.929). Our study hereby confirms previously reported cohorts, with polypharmacy as a risk factor for potential DDI in the NICU (OR: 1.60; *p* < 0.01) and PICU (≥11 prescribed medicines; *p* < 0.001) [7,25]. Similar to our study, polypharmacy (OR: 4.8) and respiratory system drugs (OR: 3.8) were the main risk factors associated with an increased incidence of DDIs in children with respiratory disorders [32].

According to a study in which the DIPS, which was also used in our study, was used in cardiovascular diseases, the predictive ability of probability scores showed good performance (AUC: 0.800, *p* < 0.001) [33]. In our study, the model predicting the presence of DDIs with ML algorithms showed a higher predictive ability (AUC: 0.929). In a study in which more than 74,000 DDIs from 572 different drugs in DrugBank (only theoretical information) were converted into a prediction model using deep learning techniques using protein binding, substrate, and enzyme, the accuracy and AUC were found to be, respectively, 0.885 and 0.921 [28]. In our real-life study, solely based on clinical data of newborns, accuracy and AUC values were higher (accuracy: 0.944, AUC: 0.929), although the number of patients and DDIs were lower. There are no ML-based studies in the literature that predict whether DDIs will occur during the period from hospitalization to discharge using clinical data.

It is reasonable to suggest that such prediction models could be instrumental in the evolution to precision medicine, with the identification of a subgroup of patients at high risk of DDI, instead of the alert fatigue related to an overload of automated alerts [26]. The limited duration of the study, number of patients, and absence of patient and health service (policy) heterogeneity due to the double-center study design are limitations. Due to the limited number of patients in a prospective study design, other parameters (risk category, ADRs, etc.) could not be included as output variables because it reduces the model performance.

## 5. Conclusions

To our knowledge, this is the first study in the literature to predict the presence of DDI using risk analysis and ML algorithms. Clinically significant DDIs were predicted with high performance according to risk analysis in neonates with PK and PD properties quite different from the pediatric and adult population. In this context, it is important to predict the likelihood of a DDI event as part of precision medicine and individualized treatment regimens.

## Figures and Tables

**Figure 1 jcm-11-04715-f001:**
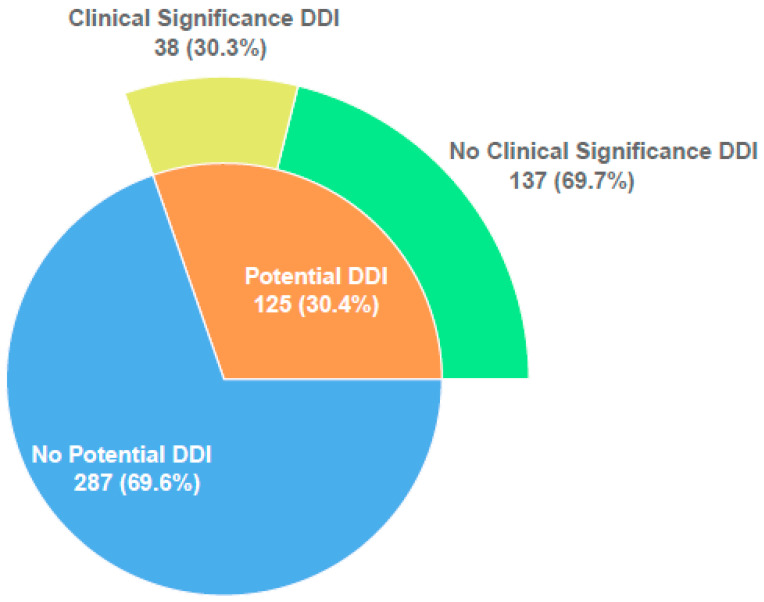
Distribution of potential and clinically relevant DDIs.

**Figure 2 jcm-11-04715-f002:**
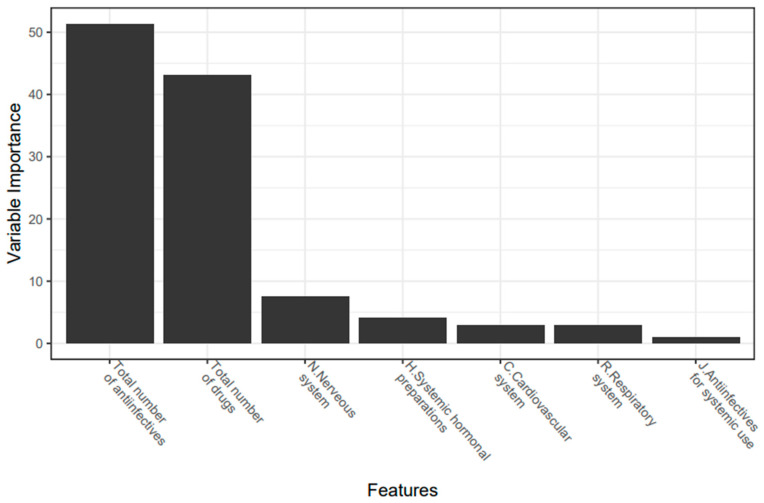
Variable importance plot (%) used to predict the presence of clinically relevant DDI.

**Figure 3 jcm-11-04715-f003:**
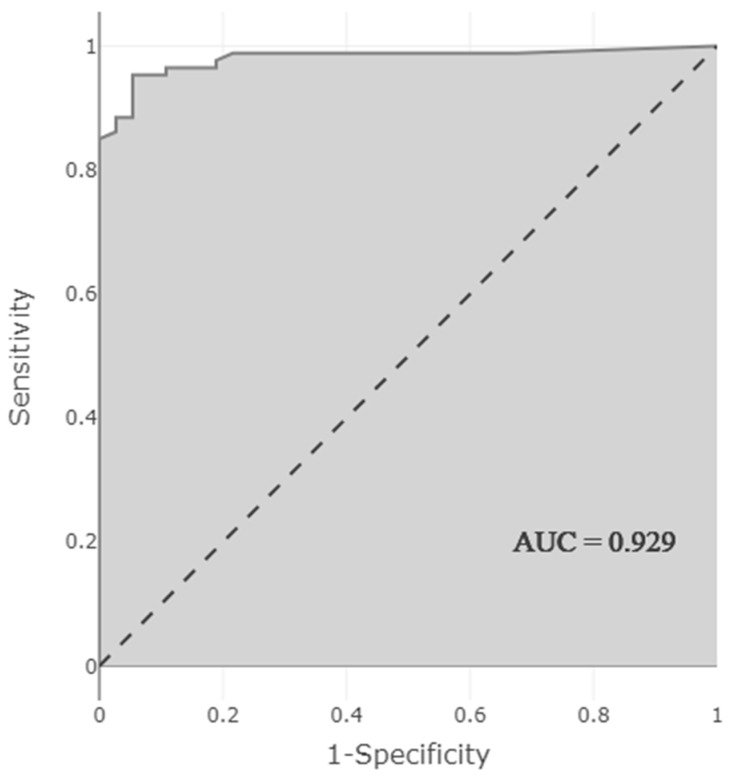
AUC-ROC curve showing the performance of the model predicting the presence of clinically relevant DDI.

**Table 1 jcm-11-04715-t001:** Data acquisition parameters of the study (N = 412).

Population Characteristics	
Sex, Male, n (%)	232 (56.3%)
Sex ratio (male/female)	1.29
5 min APGAR score, median (IQR)	8 (2)
Gestational age (weeks), median (IQR)	37 (4)
*Extremely preterm (<28 weeks), n (%)*	7 (1.7%)
*Very preterm (28 to 32 weeks), n (%)*	52 (12.6%)
*Moderate preterm (32 to 34 weeks), n (%)*	16 (3.9%)
*Late preterm (34 to 37 weeks), n (%)*	102 (24.8%)
*Term (>37 weeks), n (%)*	235 (57%)
SGA at admission, n (%)	88 (21.4%)
Birth weight (g), mean (SD)	2631.1 (877.2)
*Extremely low birth weight (<1000 g), n (%)*	26 (6.3%)
*Very low birth weight (1000 to 1500 g), n (%)*	27 (6.6%)
*Low birth weight (1500 to 2500 g), n (%)*	119 (28.9%)
*Normal birth weight (>2500 g), n (%)*	240 (58.3%)
Multiple birth, n (%)	53 (12.9%)
Caesarean section, n (%)	337 (81.8%)
Diagnosis (ICD-10), n (%)	
*Complications of labor and delivery, n (%)*	165 (40%)
*Infectious diseases, n (%)*	46 (11.2%)
*Diseases of the respiratory system, n (%)*	46 (11.2%)
*Diseases of the circulatory system, n (%)*	37 (9%)
*Other disorders of fluid, electrolyte, and acid-base balance, n (%)*	26 (6.3%)
*Diseases of the digestive system, n (%)*	24 (5.8%)
*Diseases of the nervous system, n (%)*	20 (4.9%)
*Neonatal jaundice, n (%)*	19 (4.6%)
*Congenital malformations, deformations and chromosomal abnormalities, n (%)*	15 (3.6%)
*Metabolic disorders, n (%)*	9 (2.2%)
*Neoplasms, n (%)*	6 (1.4%)
Drugs (ATC) (N = 2280), n (%)	
*J. Anti-infectives for systemic use, n (%)*	905 (39.69%)
*A. Alimentary tract and metabolism, n (%)*	591 (25.92%)
*N. Nervous system, n (%)*	229 (10.05%)
*B. Blood and blood-forming organs, n (%)*	175 (7.67%)
*C. Cardiovascular system, n (%)*	170 (7.46%)
*R. Respiratory system, n (%)*	81 (3.55%)
*H. Systemic hormonal preparations, n (%)*	70 (3.07%)
*S. Sensory organs, n (%)*	31 (1.36%)
*M. Musculo-skeletal system, n (%)*	11 (0.48%)
*G. Genito-urinary system and sex hormones, n (%)*	10 (0.44%)
*L. Antineoplastic and immunomodulating agents, n (%)*	7 (0.31%)

APGAR: Appearance, Pulse, Grimace, Activity, and Respiration, SGA: Small for gestational age, ICD: International Classification of Diseases 10th Revision, ATC: Anatomical Therapeutic Chemical.

**Table 2 jcm-11-04715-t002:** The type, outcome, duration of exposure, probability, severity, and risk category of clinically relevant DDIs observed in the study (n = 38).

Affecting Drug (Inhibitor/Inductor)	Affected Drug (Victim)	Mechanism of DDIs	ADRs Observed as a Result of DDI *	Duration of Exposure (Mean Day)	DIPS (Probability)	Lexicomp^®^ (Severity)	Risk Score	Risk Category
Vancomycin	Amikacin	Additive/synergistic	Increase in creatinine (13)	16.76	2	3	6	2
Dexmedetomidine	Fentanyl	Additive	Bradycardia (6) Hypotension (2)	3.25	2	3	6	2
Amikacin	Furosemide	Additive/synergistic	Increase in creatinine (4)	2	2	3	6	2
Dexmedetomidine	Furosemide	Additive	Hypotension (3)	4.33	2	3	6	2
Phenytoin	Phenobarbital	Metabolism	Decreased effect of phenytoin (2)	11	3	3	9	2
Hydrocortisone	Furosemide	Additive	Hypokalemia (2)	4.50	2	3	6	2
Phenobarbital	Furosemide	Unknown	Hypotension (2)	7.50	2	3	6	2
Salbutamol	Furosemide	Additive	Hypokalemia (2)	6.50	2	3	6	2
Amiodarone	Flecainide	Additive	QTc prolongation	2	4	4	16	3
Furosemide	Captopril	Volume depletion	Increase in creatinine Hypotension	4	2	3	5	2
Hydrocortisone	Furosemide	Additive	Hypokalemia	3	2	3	6	2
Hydrochlorothiazide	Diazoxide	Decrease in insulin secretion	Hyperglycemia	8	3	3	9	2
Nifedipine	Propranolol	Additive	Hypotension	5	2	3	6	2
Caffeine	Adenosine	Antagonism	Decreased effect of adenosine	10	3	4	12	3
Amiodarone	Fluconazole	Metabolism	QTc prolongation	3	2	4	8	2
Phenobarbital	Levetiracetam	Unknown	Decreased effect of levetiracetam	26	2	3	6	2
Ibuprofen	Amikacin	Unknown	Increase in creatinine	3	3	3	9	2
Spironolactone	Captopril	Increase in potassium retention due to aldosterone reduction	Hyperkalemia	18	3	3	9	2
Fluconazole	Midazolam	Metabolism	Prolonged sedation	1	2	3	6	2
Diazoxide	Dexmedetomidine	Additive	Hypotension	3	2	3	6	2
Dexamethasone	Hydrochlorothiazide	Additive	Hypokalemia	2	2	3	6	2
Fluconazole	Ibuprofen	Metabolism	Decrease in hemoglobin	2	2	3	6	2
Ciprofloxacin	Phenytoin	Unknown	Decreased phenytoin plasma concentration	5	4	2	8	2
Allopurinol	Phenytoin	Unknown	Increased phenytoin plasma concentration	1	3	3	9	2
Midazolam	Fentanyl	Additive	Chest rigidity	3	4	4	16	3
Adenosine	Dexmedetomidine	Additive	Bradycardia	3	2	3	6	2
Phenobarbital	Topiramate	Unknown	Decreased effect of topiramate	7	2	3	6	2
Phenobarbital	Dexmedetomidine	Catecholamine reduction	Hypotension	3	2	3	6	2
Fentanyl	Furosemide	Unknown	Hypotension Decreased urine output	5.50	2	3	6	2
Salbutamol	Hydrochlorothiazide	Additive	Hypokalemia	7	2	3	6	2
Phenytoin	Topiramate	Metabolism	Decreased effect of topiramate	5	2	3	6	2
Ciprofloxacin	Midazolam	Metabolism	Prolonged sedation	3	2	2	4	1
Ferrous fumarate	Levothyroxine	Absorption	Decreased effect of levothyroxine	13	2	4	8	2
Cefuroxime	Amikacin	Additive/synergistic	Increase in creatinine	1	2	3	6	2
Nitroglycerine	Furosemide	Additive	Hypotension	16	2	3	6	2
Potassium chloride	Furosemide	Unknown	Hyponatremia	1	2	2	4	1
Methylprednisolone	Furosemide	Additive	Hypokalemia	1	2	3	6	2
Linezolid	Salbutamol	Metabolism	Hypertension	10	3	4	12	3
Nitroglycerine	Dexmedetomidine	Additive	Hypotension	6	2	3	6	2
Potassium chloride	Phenobarbital	Unknown	Hyponatremia	6	2	3	6	2
Dexmedetomidine	Salbutamol	Unknown	Hypokalemia	1	2	2	4	1
Furosemide	Levothyroxine	Unknown	Increase in free T4	7	2	3	6	2
Furosemide	Levosimendan	Additive	Hypotension	1	2	3	6	2
Prednisolone	Furosemide	Additive	Hypokalemia	3	3	3	9	2
Adrenalin	Dopamine	Additive	Hypertension	3	2	3	6	2

DIPS: Drug Interaction Probability Scale, ADR: adverse drug reaction, DDI: drug–drug interaction. * The numbers in parentheses show how many times that DDI has been observed. The other ADRs were observed only once in that DDIs. Risk category column; white: low risk, light gray: moderate risk, dark gray: high risk.

**Table 3 jcm-11-04715-t003:** Distribution of potential drug–drug interactions detected by probability and severity.

**P** **R** **O** **B** **A** **B** **I** **L** **I** **T** **Y**		**SEVERİTY**				
		*A (1)* *n = 1 (0.30%)*	*B (2)* *n = 24 (7.32%)*	*C (3)* *n = 262 (79.88%)*	*D (4)* *n = 40 (12.20%)*	*X (5)* *n = 1 (0.30%)*
	*Highly Probable (4)* *n = 3 (0.91%)*	4	8	12	16	20
	*Probable (3)* *n = 16 (4.88%)*	3	6	9	12	15
	*Possible (2)* *n = 55 (16.77%)*	2	4	6	8	10
	*Doubtful (1)* *n = 254 (77.44%)*	1	2	3	4	5

1–4 points (low risk-white), 5–10 points (moderate risk-light gray), 12–20 points (high risk-dark gray).

## Data Availability

The data presented in this study are available on request from the corresponding author. The data are not publicly available due to restrictions privacy and ethical.

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
