# Peer review of "Novel Method for Early Prediction of Clinically Significant Drug–Drug Interactions with a Machine Learning Algorithm Based on Risk Matrix Analysis in the NICU"

_jcm, 2022, doi:10.3390/jcm11164715_

Round 1

Reviewer 1 Report

The paper deals with development of novel method for prediction of clinically relevant drug-drug interactions in neonates.  412 NICU patients were included in the study, who received in total 2280 drugs (about 5 drugs per patient). The most commonly used of these agents were; intravenous fluids (12.06%), gentamicin (8.03%), and ampicillin (7.81%).  328 potential drug-drug interactions were identified at 128 (30%) patients, of which 30 % were identified as clinically relevant. Most common interactions were pointed out, and the most abundant was the interaction between vancomycin and amikacin (17%), which was found to raise blood creatinine. The model for classification of DDIs by probability and severity was developed and the total number of antiinfectives and total number of drugs were identified as the most important variables. The model has shown good accuracy (95%) and fair sensitivity (0.89%). In a sample of 51 patients (of which 8 experienced DDIs), the model has correctly classified more than 90%.

The drug-drug interactions in NICU patients represent a very important and sensitive question. The presented paper is well designed and contains valuable information on the observed DDIs, their probablilty and severity. The developed model undoubtedly has limited precision due to the small sample, as admitted by auhors themselves. However, the complexity of the topics and the fact that this is the first sudy dealing with prediction of DDIs in neonates contributes to the quality of the paper.

I have one minor question:

Was the vancomycin-amikacin interaction observed in previous studies? Moreover, are there some DDIs identified in this study that were specific for neonates (not observed in older pediatric patients)? Are there some DDIs that have had more severe (or totally different, unexpected) outcome in neonates compared to older patients?

Author Response

Dear Reviewer,

Thank you for your valuable comments. Both drugs may have a potential nephrotoxic effect when used individually. If coadministered, there is a need to monitor closely for signs of nephrotoxicity. In a study, amikacin clearance was reduced by 21% and vancomcyin was reduced by 18% during co-administration of ibuprofen (1). In another small population study, although amikacin was subclinically nephrotoxic, the addition of vancomycin to amikacin therapy did not enhance clinical or tubular nephrotoxicity in the children (2).

This study did not include newborn-specific DDIs. However, due to the lack of a newborn-specific DDI database in the current literature, this web tool is the first for use in clinical practice. However, due to physiological and pharmacokinetic differences in preterm neonates, it is recommended to pay particular attention to ibuprofen-related DDIs, which are used as high dose for three days in patent ductus arteriosus (different indication) closure (Table 2). 

The severity of DDIs varies depending on gestational age, birth/admission weight, diagnosis, immaturity, procedures that cause physiological variability (etc., ECMO, CRRT), hepatic and renal function. For this reason, this web tool will give more accurate predictions because it contains neonate-specific real-life data.

References

  1. Allegaert K. The impact of ibuprofen or indomethacin on renal drug clearance in neonates. J Matern Fetal Neonatal Med. 2009;22 Suppl 3:88-91.
  2. Goren MP, Baker DK Jr, Shenep JL. Vancomycin does not enhance amikacin-induced tubular nephrotoxicity in children. Pediatr Infect Dis J. 1989 May;8(5):278-82.

Reviewer 2 Report

1.      Page 2. Line 76. What are the inclusive criteria in the study? What is the gender ratio of newborns in this study? Did the authors exclude the patients with impaired renal or liver function?

2.      Page 3. Line 92. The inhibitor/inductor may affect the same metabolic enzyme but for opposite DDI results. They are not suitable for groups in the same category.

3.      Page 5. Line 177.; Page 11. Line 222.; Page 13. Line 291. The authors used the term total number of “antiinfectives”. Page 11. Figure 2. The first column shows Total number of “antibiotics”. Please clarify whether these two terms are the same.

4.      Page 11. Figure 2. Total number of antibiotics, Total number of drugs, and J. Antiinfectives for systemic use, is it possible for these 3 categories to be counted multiple times? We did not see the number of drugs in detail.

5.      Page 13. Line 261. The authors said this is the first study to use these measurements in neonates. I would suggest the authors explain how to validate the experimental results.

Author Response

Dear Reviewer,

Thank you very much for the opportunity to submit a revised version of the manuscript " Novel Method for Early Prediction of Clinically Significant Drug-Drug Interactions with a Machine Learning Algorithm based on Risk Matrix Analysis in the NICU" (Manuscript No: JCM-1853351). We appreciate your time and consideration.

We are also grateful to the reviewers for their comments and feedback. We included practically all suggested changes in the revised version, and the manuscript has improved significantly as a result. These changes, described below, were incorporated using the track changes mode to facilitate their identification in the manuscript.

Reviewer 3 Report

The paper is generally well written and structured. The authors have developed machine learning (ML) algorithms that predict drug-drug interactions (DDI) presence by integrating each DDI. They observed at least one potential DDI in 125 (30.4%) of the neonates (n=412).  

As far as I was able to ascertain from my search of the literature,  software DDI checkers for adults are widely available. I appreciate that the authors developed the ML algorithm with the aim of predicting DDI in neonates. Because I believe that artificial intelligence applications will find more place in daily medicine practices, especially in intensive care units.

In my opinion, a high-performance web tool would be helpful for the optimization of treatment in intensive care units. The paper would contribute to the literature and can be published.

Author Response

Thank you very much for your valuable comments and feedback regarding our research paper.